# Effect of Substrate Plate Heating on the Microstructure and Properties of Selective Laser Melted Al-20Si-5Fe-3Cu-1Mg Alloy

**DOI:** 10.3390/ma14020330

**Published:** 2021-01-11

**Authors:** Pan Ma, Pengcheng Ji, Yandong Jia, Xuerong Shi, Zhishui Yu, Konda Gokuldoss Prashanth

**Affiliations:** 1School of Materials Engineering, Shanghai University of Engineering Science, Shanghai 201620, China; JiPengcheng950120@163.com (P.J.); shixuer05@mails.ucas.ac.cn (X.S.); yu_zhishui@163.com (Z.Y.); 2Laboratory for Microstructures, Institute of Materials, Shanghai University, Shanghai 200444, China; 3Department of Mechanical and Industrial Engineering, Tallinn University of Technology, Ehitajate tee 5, 19086 Tallinn, Estonia; kgprashanth@gmail.com; 4Erich Schmid Institute of Materials Science, Austrian Academy of Sciences, Jahnstraße 12, A-8700 Leoben, Austria; 5CBCMT, School of Mechanical Engineering, Vellore Institute of Technology, Vellore 632014, India

**Keywords:** Al-Si-Fe-Cu-Mg alloy, 4XXX series, selective laser melting, heat treatment, microstructure, mechanical property

## Abstract

The Al-20Si-5Fe-3Cu-1Mg alloy was fabricated using selective laser melting (SLM). The microstructure and properties of the as-prepared SLM, post-treated SLM, and SLM with substrate plate heating are studied. The as-prepared SLM sample shows a non-uniform microstructure with four different phases: fcc-αAl, eutectic Al-Si, Al_2_MgSi, and δ-Al_4_FeSi_2_. With thermal treatment, the phases become coarser and the δ-Al_4_FeSi_2_ phase transforms partially to β-Al_5_FeSi. The sample produced with SLM substrate plate heating shows a relatively uniform microstructure without a distinct difference between hatch overlaps and track cores. Room temperature compression test results show that an as-prepared SLM sample reaches a maximum strength (862 MPa) compared to the heat-treated (524 MPa) and substrate plate heated samples (474 MPa) due to the presence of fine microstructure and the internal stresses. The reduction in strength of the sample produced with substrate plate heating is due to the coarsening of the microstructure, but the plastic deformation shows an improvement (20%). The present observations suggest that substrate plate heating can be effectively employed not only to minimize the internal stresses (by impacting the cooling rate of the process) but can also be used to modulate the mechanical properties in a controlled fashion.

## 1. Introduction

Aluminum-silicon alloys, especially 10–20 wt% Si alloys, are widely used in automotive parts, cylinders, cylinder heads, brakes and other automotive parts due to their light weight, high specific strength and good wear resistance [1,2,3,4,5]. Elements such as Cu, Cr, Mg, Fe, and Ni are normally added to Al-Si hypereutectic alloys to improve their strength, wear resistance, and thermal stability [6,7,8,9,10,11]. In particular, the addition of Fe provides better chemical stability and thermal stability [12,13,14]. However, in fabrication of alloys with higher iron content, the conventional casting process results in coarse grains as a result of relatively lower cooling rate [15]. The coarse-grained structure act as sites for crack initiation and deterioration of mechanical properties [16,17,18,19,20]. It is well known that rapid solidification can shorten the diffusion time of an element, thereby reducing the size of the Si phase, and the modification of the Si phase can also be achieved [20,21,22,23,24]. Laser-based powder bed fusion (LPBF) or selective laser melting (SLM) is one of the emerging powder bed fusion technologies [25,26,27] that has been developed rapidly in recent years and has shown great potential in improving the mechanical properties and can fabricate components with a complex geometry in Al-Si alloys [28,29,30,31]. Moreover, the freedom design and high material usage efficiency make this technique unique and has gained favorable attention [32,33,34,35].

Generally, Al and its alloys are known for their high thermal conductivity (thermal conductivity represents the ability of the materials to conduct the heat) and thermal diffusivity (the rate of heat transfer in materials is characterized by the thermal diffusivity). It suggests that the Al-based alloys dissipate heat at very high rates. Hence, the powder bed not only reflects the laser beam away, but also conducts the heat rapidly away from the melt pool into the previously solidified layer and to the powder bed surrounding the melt pool or the substrate plate. Hence, it requires an additional supply of energy to compensate for the rapid conduction of heat from the melt pool. Such high energy supplied to the powder bed will result in a significantly wider melt pool that creates limitations in the minimum size of the features that can be produced by the SLM process. In addition, high heating and cooling rates that lead to significant thermal gradients around the melt pool may cause undesirable levels of cracking. In general, the problems associated with the fabrication of Al and its alloys by SLM are: (a) oxidation of the surface of the metal powder; (b) obstructed flowability of the powder; (c) low absorptivity of the laser beam and high reflectivity; (d) high thermal conductivity and, hence, wider melt pools placing restrictions on the size of the smallest features in the part that can be fabricated; (e) high solidification shrinkage (may lead to cracking); and (f) high viscosity of the melt [36]. Such difficulties may result in undesirable microstructures resulting in poor properties of SLM parts. In addition, Al and its alloys may face other problems such as (a) porosity—improper processing parameters; (b) balling due to too high-energy input; (c) formation of a distorted layer due to too high-energy input and/or with the presence of brittle parts; (d) increased cracking tendency due to the brittle nature of the processing material; (e) high surface roughness of parts because of coarse powder and/or too high energy input; (f) loss of alloying elements, especially when the alloy contains elements with low boiling points and vapor pressures that can be lost during the SLM process; and (g) poor dimensional accuracy due to the presence of oxide layers, where the energy input has to be increased unprecedentedly [36,37,38].

To mitigate some of these issues, recently, effects have been made to fabricate Al-Si alloy from elemental additions instead of using pre-alloyed powders by using additive manufacturing technology. Xiong et al. [39] studied the effect of cellular structure and melt pool boundary condition on the mechanical properties, deformation, and failure behavior of AlSi10Mg alloy processed by SLM. Similarly, Pozdniakov et al. [40] have studied the microstructure and mechanical properties of the novel AlSi11CuMn alloy manufactured by SLM and Wang et al. [41] fabricated a heat-treatable Al-3.5Cu-1.5Mg-1Si alloy, and the corresponding microstructures and mechanical properties were investigated. In addition, Wang et al. have showed successful fabrication of Al-Cu material and Zhao et al. have showed the successful fabrication of Cu-Ni-Sn samples from elemental powders using SLM [42,43]. These reports show that not only elemental powders may be used for fabricating parts by SLM but also the parts with a refined microstructure may be obtained due to high cooling rates observed during the SLM process [44,45,46].

Selective laser melting (SLM) generates large thermal gradients in the melt pool. As a result, a mismatch in elastic deformation is observed, which leads to high levels of residual stress within the additively manufactured metallic structure [47]. Currently, the effective method to minimize such residual stresses is to employ substrate plate heating (in situ) or an external post heat treatment (ex situ) has to be carried out [48,49,50,51,52]. Generally, samples by SLM are fabricated over a base plate/substrate plate. The heat extraction from the melt pool happens through the substrate plate, which is solid material made of the same composition as the powder or closest matching composition. Hence, the physical properties of the solidified melt pool and the substrate plate are similar. Accordingly, in this study, Al-Si based heat treatable alloy with addition of transition metals like Fe and Cu along with Mg is chosen. The addition of Fe promotes the formation of AlSiFe-based phase and the addition of Mg suppresses the formation of Al-Cu precipitate but promotes the formation of AlCuMg based phase. These phases are expected to enhance the strength of the Al-based material. However, at the same time, these intermetallic phases are rather brittle compared to the Al-matrix and hence introduce additional internal stresses, when produced by SLM [11]. When the local internal stresses exceed the local fracture stress, it may lead to cracking in the material. Hence, the cooling rate of the process should be carefully reduced/controlled to fabricate such materials by SLM. In this study, both substrate plate heating and post heat treatment methods were adopted and the effects of substrate plate heating and post heat treatment on the microstructure, phase composition, and mechanical properties of selective laser melted Al-20Si-5Fe-3Cu-1Mg alloys were studied systematically and in detail.

## 2. Materials and Methods

Spherical gas atomized Al-20Si-5Fe-3Cu-1Mg powders with particle size distribution of 20–63 µm was used as the raw materials for the SLM process. The chemical composition (measured using optical emission spectroscopy—OES method) is listed in Table 1. An SLM Solutions 280 HL (SLM Solutions Group AG, Lubeck, Germany) machine equipped with a Yb-YAG fiber laser was used in the experiment. Prior to the experiments, the build chamber was flooded with argon gas to reduce the oxygen level to less than 0.02 ppm, and the argon flow (2.5 L/min) was maintained throughout the experiment. Argon was blown from the right side of the build chamber towards the left and the directionality is maintained throughout the fabrication process.

Other parameters set in the SLM build processor include: laser power of 320 W, layer thickness of 50 μm, laser scan speed of 1455 mm/s, hatch space of 110 μm, and the scanning direction was rotated 67° between consecutive layers to obtain optimal densification. The SLM samples were then heat-treated to 673 K for 6 h in an argon-controlled atmosphere. In addition, additional samples were produced with powder bed heating, where the substrate plate was heated to 673 K throughout the fabrication process. Compression samples of 8 mm in height and 3.5 mm in diameter were prepared for the experiments. The samples are built perpendicular to the substrate plate (standing).

Light microscope (LM) (Olympus, Hamburg, Germany) and scanning electron microscope (SEM) Gemini 1530 microscope (Jeol, München, Germany) operating at 20 kV were used to study the microstructure of samples fabricated with different processing conditions. The samples that were used for microstructure study were polished using conventional metallography methods. Polished surfaces were then etched for 30 s using 0.5% HF solution (0.5 HF, 99.5 H_2_O, in vol%). The structural information was extracted from the diffraction patterns that were obtained by X-ray diffraction (XRD) using a PANalytical X’Pert Pro Diffractmeter (PANalytical, Almelo, The Netherlands) with Co-Kα radiation (*λ* = 1.7902 Å) in reflection geometry. The samples were scanned with a step size of 0.02°/s at ambient conditions and in the 2*θ* range 20° and 110°. The crystallite size (*D*) of the samples were calculated using the Scherrer Equation [53],(1)$$D=\frac{K\times \lambda }{B_{2\theta }\times cos\theta },$$
where *K* is a constant, generally assumed to be 0.9, *λ* is the wavelength of the X-ray used, *B*_2*θ*_ is the full width and half maximum of the diffraction peak, and *θ* is the diffraction angle.

The dislocation density for the SLM samples were calculated from the XRD patterns using the following methodology: the strain as well as the crystallite size were calculate using the Williamson–Hall equation [54,55]. Once the internal strain and crystallite size were calculated, the dislocation density was calculated using the Equation,(2)$$\rho_{d}=\frac{2\sqrt{3}{\langle \varepsilon^{2}\rangle }^{0.5}}{bd},$$
where, *ρ_d_* is the dislocation density, *ε* is the internal lattice strain, *b* corresponds to the magnitude of the burgers vector, and *d* represents the crystallite size. The microstructure of these SLM fabricated and heat-treated materials were also examined with a Philips CM12 transmission electron microscopy (TEM) (Philips, Hamburg, Germany) equipped with Energy Dispersive X-Ray Spectroscopy (EDS) operating at 120 kV. TEM disc-shaped specimens were prepared by using ion-beam thinning at a current of 0.5 mA and an inclination angle of 15°. Vickers microhardness tests were conducted using a 1.9613 N load on a Zwick microhardness tester (Schimadzu, Dusiburg, Germany). An average hardness value was calculated for each sample using 12 indentations. The microhardness measurements were made on the cross-section of the sample, especially on the top of the as-prepared specimen (approximately 7.5 mm from the substrate plate). The hardness measurements on the samples are always made at similar heights from the substrate plate, so the anisotropy in microstructure and mechanical properties in an additively manufactured sample especially using substrate plate heating can be mitigated. The compression tests were carried out according to the standard cylindrical specimens of 3 mm diameter and 6 mm height (aspect ratio of 2). The SLM samples were polished to a diameter of 3 mm (from the original diameter of 3.5 mm) using polishing papers (GRIT 1000, 2500, and 4000) to have a smooth surface. The height of the samples was reduced to 6 mm (from the original height of 8 mm) and the plane parallel condition between the two surfaces of the sample is confirmed before the compression tests to avoid loading errors. In addition, it was made sure that the samples were always tested in such a way that the bottom of the as-built sample forms the bottom of the compression sample so that the errors that may occur due to orientation differences are avoided. At least three samples were tested for each condition using an INSTRON 5569 testing machine (INSTRON GmbH, Darmstadt, Germany) at a strain rate of 10^−4^ s^−1^.

Room temperature nanoindentation tests were performed using Agilent Nano-indenter G200 (Agilent, Waldbronn, Germany) with a standard three-sided Berkovich diamond. The nano-indentation experiments were carried out using a load-controlled mode and the maximum load of the indentation was always controlled at around 8 mN for all cases. The area function of the indenter had been calibrated on the standard fused silicon before nano-indentation tests. The thermal drift was calibrated to be less than 0.05 nm/s prior to indentation tests. During the indentation tests, the Berkovich diamond indenter (tip radius 100 nm) was loaded at the rate of 1.6 mN/s to a peak load of 8 mN with a holding time of 30 s. The surface of Al-20Si-5Fe-3Cu-1Mg alloy was mechanically polished to mirror finishing to acquire reliable data. Hardness mapping was plotted using nanoindentation data conducted with a fixed distance of 15 μm between two indents. A matrix of 8 × 8 is used for plotting the hardness maps. Similar to the microhardness measurement, the nanoindentation maps were carried out on the as-prepared samples along the cross-section at a distance approximately 7.5 mm from the substrate plate.

## 3. Results and Discussion

Figure 1 shows the microstructure of the SLM Al-20Si-5Fe-3Cu-1Mg alloy parallel to the plane of base plate (cross-section). The low magnification image in Figure 1a indicates that the microstructure of the SLM sample is non-homogenous and displays typical features like the laser tracks including hatch overlaps and track cores, characteristic for the SLM fabricated samples [11]. The morphology of the phases observed in the hatch overlaps and cores have distinct differences and are examined separately in Figure 1b,c, respectively. The microstructure in the hatch overlaps (Figure 1b) is composed of primary Si phase (particles in the size of 2 ± 1 µm), Al-Si eutectic (eutectic Si is observed as dendritic shape in the sub-micron/nano regime with the size of 0.7 ± 0.3 µm), Al-Si-Fe phase (as platelets with an average length of 11 ± 1 µm and width of 0.8 ± 0.3 µm), and Al_2_CuMg phase (as particles in the size 0.4 ± 0.1 µm). The Al_2_CuMg phase is randomly distributed in the Al-matrix, which is found on the surface of the Al-Si-Fe compound. The Al-Si-Fe compound forms prior to the eutectic reaction, and provides nucleation sites for eutectic Si phase, and most eutectic Si phase grow around the Al-Si-Fe phase. The phases observed in the track cores (Figure 1b) is identical to the phases observed in the hatch overlaps, excepting that the morphology of the phases is much finer in the track cores. This is because in the hatch overlaps, the material is melted twice and hence are exposed to slower cooling rates than in the track cores. Hence coarser microstructure is observed in the hatch overlaps and finer microstructure in the track cores like most of the SLM fabricated samples [56,57]. The size of the Al-Si-Fe phase significantly decreases from 11 ± 1 µm to 2.5 ± 0.5 µm. Moreover, the eutectic Si changes its morphology from dendrite to particle shape.

Figure 2 exhibits the microstructure of SLM Al-20Si-5Fe-3Cu-1Mg alloy after heat treatment at 673 K for 6 h. As observed from Figure 2a, the microstructure is still anisotropic with distinct differences observed in the morphology of the phases observed in both the hatch overlaps and the track cores. Figure 2b shows the higher magnification image of the hatch overlaps. The phases Al-Si-Fe (length: 11 ± 1 µm and width 0.6 ± 0.3 µm) and primary Si phase (2 ± 1 µm) show similar size like the as-SLM sample without any distinct changes in them even after thermal treatment at 673 K for 6 h. However, the eutectic Si changes its morphology from dendrite shape to particulate, and its size increases to about 1.2 ± 0.3 μm. Similarly, the morphology of the track cores shown in Figure 2c, exhibits that the edges of both primary Si and eutectic Si disappear and become particulate shape. The size of the Al-Si-Fe phase is similar as SLM condition (2.5 ± 0.6 µm) (Figure 1c).

The microstructure of the SLM Al-20Si-5Fe-3Cu-1Mg alloy built with substrate plate heating (Figure 3) shows a relatively homogeneous microstructure. Both the hatch overlaps and track cores show nearly uniform microstructure and it is hard to distinguish the microstructural features between them. The microstructures show that the Al-Si-Fe phase displays a long rod-like (acicular) morphology with length of about 15 ± 0.5 μm and width of about 1 ± 0.3 µm. The primary Si phase shows polygon shaped morphology with a size of about 5 ± 0.5 μm. The eutectic Si exhibits a particulate shape with an average size of about 1.5 ± 0.3 μm. As the powder bed is constantly held at 673 K during the entire fabrication process, the morphology of the phases has shown a significant growth in them, especially the primary Si phase. These variations in the microstructure is direct representation of the cooling rate changes. Since the samples produced with the substrate plate heating show less thermal gradients due to reduced cooling rates exhibited during the process [52]. Hence, adequate time is available during the cooling process for the primary Si phase to grow. In addition, the substrate plate heating increases the stability of the melt pool, which can help in avoiding the undesirable effect like splashing and Marangoni phenomena. Moreover, the heat introduced to the powder bed through the substrate plate also contributed to the overall energy density, which not only helps in better consolidation of the powder, but also leads to a relatively uniform microstructure.

The XRD patterns of the Al-20Si-5Fe-3Cu-1Mg alloy fabricated with different conditions are furnished in Figure 4. The diffraction pattern of the as-prepared SLM sample shows the presence of α-Al (fcc) and Si (fcc) peaks corresponding to the Al-Si eutectic and primary Si phases. The lattice parameter of Al and Si phases are observed to be 0.40497 nm and 0.54302 nm, respectively. The lattice parameter of Al suggests that it does not form any super saturated solid solution unlike in the case of Al-12Si, Al-20Si, or Al-50Si samples fabricated by SLM [48,58,59]. However, the intensity of the first two intense peaks of the Al phase show reversal of their peak intensities confirming the presence of crystallographic texture in the sample. It has been observed in most of the Al-xSi processed materials that the preferred growth is prevalent and it does not influence the mechanical properties significantly and can their presence can be ignored [48]. In addition, peaks of two additional intermetallic phases are observed namely: Al_2_MgCu and δ-Al_4_FeSi_2_, which are the most common phases that are observed in the Al-Cu-Mg-Si and Al-Fe-Si based systems. Al_2_MgCu phase is generally termed as S phase, which exhibits an orthorhombic crystal structure (Cmcm space group) with the following lattice parameters: a = 0.400 nm, b = 0.923 nm, and c = 0.714 nm respectively. On the other hand, the δ-Al_4_FeSi_2_ phase belongs to tetragonal family with the following lattice parameters: a = 0.609 nm and c = 0.944 nm. The average dislocation density observed in the as-prepared SLM sample is found to be 4 × 10^15^ m/m^3^, which is two orders of magnitude higher than the conventional cast/powder metallurgical samples or the gas-atomized powder. The crystallite size of the Al phase is found to be ~108 ± 4 nm.

In general, the δ-Al_4_FeSi_2_ phase is a meta-stable phase and it normally transforms to β-Al_5_FeSi phase below 883 K and at equilibrium cooling rate. Hence, supply of thermal energy is expected to cause a phase transformation to achieve stable phases from the meta-stable phase. The XRD pattern of the SLM sample after thermal treatment at 673 K for 6 h has shown the presence of the following phases: α-Al, Si, Al_2_MgCu, and δ-Al_4_FeSi_2_ like the as-prepared SLM sample. The lattice parameters of α-Al and Si phases are observed to be 0.40495 nm and 0.54300 nm, respectively. However, the texture present in the Al-phase is partially reversed with the thermal treatment. The peaks become relatively narrow, since the internal stress in the samples are relaxed during the thermal treatment. In addition, the dislocations also get annihilated during the thermal treatment and the dislocation density is reduced to 7 × 10^14^ m/m^3^. The crystallite size of the Al phase in the thermally treated sample is found to be ~220 ± 5 nm, which is nearly double the crystallite size of the Al phase observed for the as-prepared SLM sample. Moreover, additional peaks of β-Al_5_FeSi is observed, which should come from the partial transformation of the meta-stable δ-Al_4_FeSi_2_ during the thermal treatment. The meta-stable acicular tetragonal δ-Al_4_FeSi_2_ phase transforms to monoclinic β-Al_5_FeSi phase with the following lattice parameters: a = b = 0.612 nm and c = 0.4150 nm (β = 91°). The results corroborate with the SEM images, where the needle-like Al-Si-Fe phase is found to be a mixture of δ-Al_4_FeSi_2_ and β-Al_5_FeSi compounds in the SLM sample after heat treatment at 673 K for 6 h. It was evident from the XRD and SEM images that the meta-stable tetragonal δ-Al_4_FeSi_2_ transform to monoclinic β-Al_5_FeSi phase; however, the transformation is not complete, hence showing the presence of two different Al-Fe-Si based phases simultaneously. The intensity of the peaks in the heat treated SLM sample corresponding to the δ-Al_4_FeSi_2_ phase decreases relatively to the as-prepared SLM sample, which is consumed by the formation of monoclinic β-Al_5_FeSi phase.

During solidification, the primary Si precipitates first, and then the metastable δ-Al_4_FeSi_2_ should form [60]. Once the nucleation of δ-Al_4_FeSi_2_ phase takes place, both Fe and Si elements diffuse to the edge of δ-Al_4_FeSi_2_ phase because of the decreasing solubility of these two elements in α-Al matrix [61], promoting further growth of δ-Al_4_FeSi_2_ phase with one fast growth direction and two slow directions, which contributes the needle-like morphology. Subsequently, δ-Al_4_FeSi_2_ phase will be transformed into β-Al_5_FeSi phase [62], which can be accomplished via the diffusion of Fe and Si atoms from δ-Al_4_FeSi_2_ phase to α-Al matrix. However, low diffusion coefficients of Fe and Si atoms in α-Al matrix (D_Fe_ = 7.923 × 10^−14^ m^2^/s and DSi = 1.524 × 10^−12^ m^2^/s at 883 K, calculated according to [63]). In addition, the high cooling rate observed during the SLM solidification process both limit the diffusion of Fe and Si atoms, especially, the plentiful diffusion of Si atom from δ-Al_4_FeSi_2_ phase with high concentration to α-Al matrix. As a result, δ-Al_4_FeSi_2_→β-Al_5_FeSi transformation cannot be carried out so that the δ-Al_4_FeSi_2_ phase becomes the dominant Fe-bearing phase in SLMed alloy. However, with additional thermal treatment, the transformation of δ-Al_4_FeSi_2_→β-Al_5_FeSi is realized because of the slow heating and cooling rates associated with the thermal treatment.

The XRD pattern of the sample produced by SLM with substrate plate heating of 673 K shows similar phases like the heat-treated SLM sample. The following phases are indexed: α-Al, Si, Al_2_CuMg, δ-Al_4_FeSi_2_, and β-Al_5_FeSi, very similar to the heat-treated SLM sample, excepting some changes in the intensity of some peaks. The intensity of the β-Al_5_FeSi phase increases at the cost of δ-Al_4_FeSi_2_ when compared to the heat-treated SLM sample. In addition, the texture in the Al phase observed in the as-SLM sample is completely absent in the sample fabricated with substrate plate heating. The XRD patterns of the top and bottom (not shown here) of the samples were examined closely, where no distinct differences were observed, except that the peaks were slightly broader in the top of the sample than the bottom. This is because of the differences in the crystallite size, internal stresses, and/or dislocation density. The crystallite size in the top part of the SLM sample produced with substrate plate is found to be 125 ± 10 nm and in the bottom part is found to be 175 ± 10 nm. The top part of the sample shows a crystallite size close the as-prepared SLM sample (108 ± 4 nm) and the crystallite size at the bottom of the sample approaches the 673 K, 6 h thermally treated sample (220 ± 5 nm). Similarly, the dislocation density in the top part of the sample was found to be 3 × 10^15^ m/m^3^ and in the bottom of the sample, it is found to be 5 × 10^14^ m/m^3^. Hence, there is a strong anisotropy in terms of defects and microstructure along the length of the sample. When solidified with a relatively low cooling rate (the alloy solidified with substrate plate heating), the diffusion time of Fe and Si atoms from δ-Al_4_FeSi_2_ phase to α-Al matrix would increase and partial δ-Al_4_FeSi_2_ phase would transform to β-Al_5_FeSi phase and Si phase. If transformation could not complete in the δ phase particles due to lack of time, some of the plate-shaped particles will end up with δ phase in the core and the β phase on the shell/outside, as given in Figure 5.

However, the Al-20Si-5Fe-3Cu-1Mg gas atomized powder [64] shows the presence of only three distinct phases namely: α-Al, Si, and δ-Al4FeSi2 as compared to the five different phases (Al, Si, Al2CuMg, δ-Al4FeSi2, and β-Al5FeSi) observed after the SLM process. In addition, other compositions related to the present composition (Al-20Si-5Fe-3Cu-1Mg) like Al-20Si-5Fe [65,66] and Al-20Si-5Fe-2X (X = Cu, Ni, Cr) [10] irrespective of the form present (powder or melt-spun ribbons or hot consolidated specimens) show the presence of three phases (α-Al, Si, and δ-Al_4_FeSi_2_) like the gas atomized Al-20Si-5Fe-3Cu-1Mg powder. Hence, in the present case, the non-equilibrium nature of the SLM process and the presence of anisotropic heat extraction conditions [67] lead to the formation of the other two phases, Al_2_CuMg and β-Al_5_FeSi.

The Vickers microhardness values of the Al-20Si-5Fe-3Cu-1Mg alloy in the as-prepared SLM, SLM + heat treated and SLM + substrate plate heating conditions are demonstrated in Figure 6. The as-prepared SLM sample exhibits the highest hardness value of about 240 ± 5 HV_0.1_, which is representative of the manufacturing process. The Vickers microhardness of the Al-20Si-5Fe-3Cu-1Mg gas atomized powder is observed to be 234 HV, which is very close to the hardness of the SLM sample, suggesting that SLM and gas atomization process exhibit similar cooling rates, however different is the solidification conditions [64]. However, similar compositions like Al-20Si-5Fe, and Al-20Si-5Fe-2X (X = Cu, Ni, Cr) [10,66,67] processed via powder metallurgical route (gas atomization and subsequent hot consolidation) show hardness of the samples varying in the range 140–200 HV. The hardness is significantly lower than the present SLM sample owing to the presence of Cu and Mg, which promotes the formation of the intermetallic phases Al_2_CuMg and β-Al_5_FeSi. The hardness value of the SLM alloy after heat treatment at 673 K for 6 h is 166 ± 7 HV_0.1_ and the value decreases to 142 ± 4 HV_0.1_ for the alloy produced by SLM substrate plate heating at 673 K. As discussed above, due to the high cooling rates observed during the SLM process, the SLM alloy displays fine microstructure and hence higher hardness.

During thermal treatment, the phases grow in terms of grain size and hence due to such coarsening, the strength of the materials show a drop, irrespective of external heat treatment or substrate plate heating. The difference in hardness observed between the sample produced with based plate heating and the thermally treatment materials is less than 20 HV_0.1_ and hence they are not so significant. The decrease in the hardness may also be attributed using the Hall–Petch relationship, where increase in the size of the phases decreases the hardness/strength of the material and vice-versa. In addition, the residual stress in the as-prepared SLM sample also decreases or nearly eliminated. Similarly, the sample produced with substrate plate heating exhibits less thermal gradients in them, less degree of internal stresses and possibly lower amount of internal defects. Since the sample is exposed to 673 K through the substrate plate for the entire duration of the build, the sample is constantly exposed to high temperatures and as a result, the cooling rate of the SLM process is decreased drastically. Hence, the sample produced with substrate plate heating show a decreased hardness compared to the SLM sample.

Nanoindentation experiments were used to plot the hardness distribution map for all the three sample conditions considered. The average nano-hardness observed for the samples are 3.4 ± 0.5 GPa, 1.9 ± 0.5 GPa, and 1.8 ± 0.5 GPa in as-prepared SLM, heat treated SLM, and SLM sample prepared with substrate plate heating, respectively. The hardness results corroborate with the microhardness data shown in Figure 6. It can be observed from all the three plots in Figure 7 that the distribution of hardness is not uniform and shows anisotropy throughout the surface of the sample. This anisotropy in the hardness distribution may be attributed to the non-uniform distribution of the phases in the hatch overlaps and track cores, where regions of coarse morphology is found along hatch overlaps and relatively finer morphology along track cores. Hence in the as-prepared SLM sample, the hardness varied between 4.1 GPa to 2.5 GPa (a difference of 1.6 GPa) depending on the hatch overlaps and track cores and also on the type of the phases present. For instance, the fcc-Al phase should exhibit low hardness, whereas the intermetallic phases should exhibit high hardness values and hence these exists a local anisotropy in hardness distribution. In the heat-treated SLM sample, the hardness varies between 2.9 GPa and 1.1 GPa (a difference of 1.8 GPa). Such a huge difference also comes from (1) non-uniform distribution of phases and (2) difference in morphology of the phases between hatch overlaps and track cores, which is similar to the as-SLM condition, excepting for lower hardness levels due to the coarsening and grain growth that usually takes place with the supply of thermal energy. Figure 7c shows the hardness distribution plot for the SLM sample prepared with substrate plate heating. The hardness values vary between 4.6 GPa and 0.7 GPa (a difference of 2.9 GPa). Such huge variation in hardness can be attributed to the variation in the microstructure along the hatch overlaps and track cores, presence of five different phases (fcc Al, Al-Si eutectic, Al_2_MgCu, δ-Al_4_FeSi_2_, and β-Al_5_FeSi phase), where fcc-Al is quite soft and the intermetallic phases are rather hard and brittle. In addition, the substrate plate heating also causes anisotropy in the sample along the direction of the build and hence the hardness shows an anisotropic variation along the building length. Considering all the three sample conditions the as-prepared SLM sample is less anisotropic than the sample SLM sample fabricated with base plate heating. Any thermal treatment offered reduces the anisotropy in the material, which can be observed from the SLM + heat-treated sample along with the strength of the material (Figure 7b).

Representative room temperature compressive stress-strain curves in quasistatic mode is presented in Figure 8. The highest strength of 862 ± 5 MPa (ultimate compressive strength) is observed for the as-prepared SLM sample at the expense of compressive deformation. Here the soft-fcc Al phase is accompanied by fine eutectic Al-Si other intermetallic phases in the form of particles or platelets (Al_2_MgCu, δ-Al_4_FeSi_2_, and β-Al_5_FeSi). The SLM processed material has a high density of dislocations that is two orders of magnitude higher than the conventionally cast or powder metallurgical counterparts. Dislocation movement is associated with the plastic deformation and during compression of the samples, the dislocations tend to move. This movement of the dislocation is confined along the grain boundaries (according to the confinement theory), which leads to strengthening of the material. In addition, the hard and brittle intermetallic phases act as a barrier for the crack propagation (crack arresters) thereby further increasing the strength of the material under compression. When the SLM samples are thermally treated to 673 K for 6 h, the ultimate compressive strength of the material reduces to 524 ± 6 MPa, but the compressive deformation increases to ~17%. The sample produced by SLM with substrate plate heating shows an ultimate compressive strength to 474 MPa and the deformability to ~20%. Such high compressive strengths can be ascribed to the fine microstructure along with presence of intermetallic phases and the Al-Si eutectic phase. The reduction in the mechanical strength/hardness in the heat treated as well as the samples produced with substrate plate heating are attributed to the presence of a reduced amount of residual stresses and grain growth due to availability of sufficient thermal energy.

The results demonstrate that the substrate plate heating not only reduces the strength of the material, but also improves the plastic deformability of the material, due to the reduction of cooling rate with the employment of substrate plate heating. The reduction in the cooling rate helps in the formation of relatively coarse material with less degree on internal stresses compared to its counterpart produced without substrate plate heating. In addition, the substrate plate heating also reduces the thermal gradient in the melt pool during solidification. All these effects lead to significant changes in the microstructure and hence, the fracture strain under room temperature compression increases to about 18% and the ultimate strength decreases accordingly. Therefore, employment of substrate plate heating is not only considered to reduce the internal stresses in the material, but it also has a distinct impact on the microstructure and their mechanical properties, which can be altered in a controlled manner (depending on the substrate plate heating temperature).

## 4. Conclusions

The microstructure, phase composition, and mechanical properties of the as produced, heat-treated, and SLM with base plate heated Al-20Si-5Fe-3Cu-1Mg alloy have been investigated in detail. The following conclusions can be drawn:(1)The microstructure of SLM built sample and the heat-treated sample show non-uniform distribution of phases with coarse features observed in the hatch overlaps and fine features along the track cores.(2)The sample fabricated with SLM with base plate heating do not show typical hatch, core strategy. However, the sizes of the phases are relatively larger compared the as-prepared SLM samples.(3)The needle-like Al-Si-Fe phase observed in SLM sample is composed of δ-Al_4_FeSi_2_ phase, whereas it is a mixture of δ-Al_4_FeSi_2_ and β-Al_5_FeSi compounds in both the post heat-treated and SLM base plate heated Al-20Si-5Fe-3Cu-1Mg samples.(4)The alloy fabricated with SLM exhibits the highest hardness value (~240.25 HV_0.1_) and highest compression strength (~862 MPa). The SLM alloy with substrate plate heating at 673 K displays the lowest strength and the largest plastic deformation. The ultimate compression strength of the as-prepared SLM, the post heat-treated, and the SLM base plate heated samples are observed to be 862 MPa to 524 MPa and 474 MPa, respectively. Correspondingly, the plastic deformation increases from 7% to 17% and 20%, respectively, for these samples.(5)Similarly, the average nano-hardness of as-prepared SLM alloy is ~3.4 ± 0.5 GPa, and are ~1.9 ± 0.5 GPa and 1.8 ± 0.5 GPa, respectively for post heat treated SLM and SLM base plate heated samples.

The results suggest that Al-Si-TM based alloys can be fabricated by SLM process to achieve high compression strength and at the same time the strength and plastic deformation of the material can be tuned or modulated by carefully controlling the cooling rate (by employing base/substrate plate heating with desired temperatures) in a controlled manner. Nevertheless, the samples, when produced with substrate plate heating will have an anisotropy along the length of the samples, which may not be desired.

## Figures and Tables

**Figure 1 materials-14-00330-f001:**
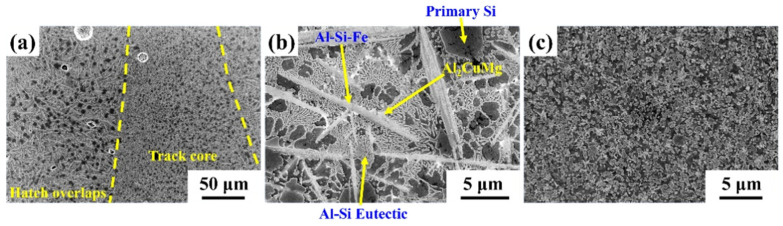
Scanning electron microscopy images of the SLM fabricated Al-20Si-5Fe-3Cu-1Mg alloy: (**a**) low magnification image showing the presence of hatch overlaps and track cores, (**b**) hatch overlaps showing the phases: primary Si, Al-Si-Fe, Al_2_CuMg, and Al-Si eutectic, and (**c**) track cores showing fine morphology of the phases.

**Figure 2 materials-14-00330-f002:**
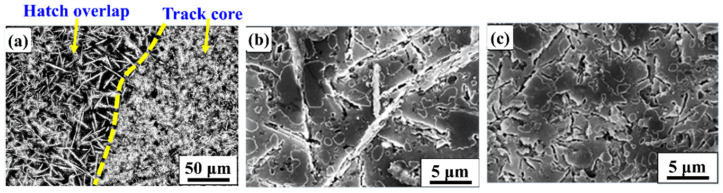
Images showing the scanning electron microstructures of the SLM fabricated Al-20Si-5Fe-3Cu-1Mg alloy after heat treatment at 673 K for 6 h: (**a**) low magnification image showing both hatch overlaps and track cores, (**b**) higher magnification image showing the hatch overlaps, and (**c**) higher magnification images showing track cores with fine features compared to the hatch overlaps.

**Figure 3 materials-14-00330-f003:**
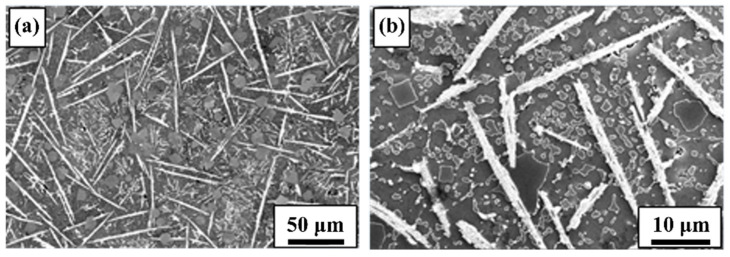
Scanning electron microscopy images of the SLM fabricated Al-20Si-5Fe-3Cu-1Mg alloy with substrate plate heating to 673 K: (**a**) low magnification image and (**b**) high magnification image.

**Figure 4 materials-14-00330-f004:**
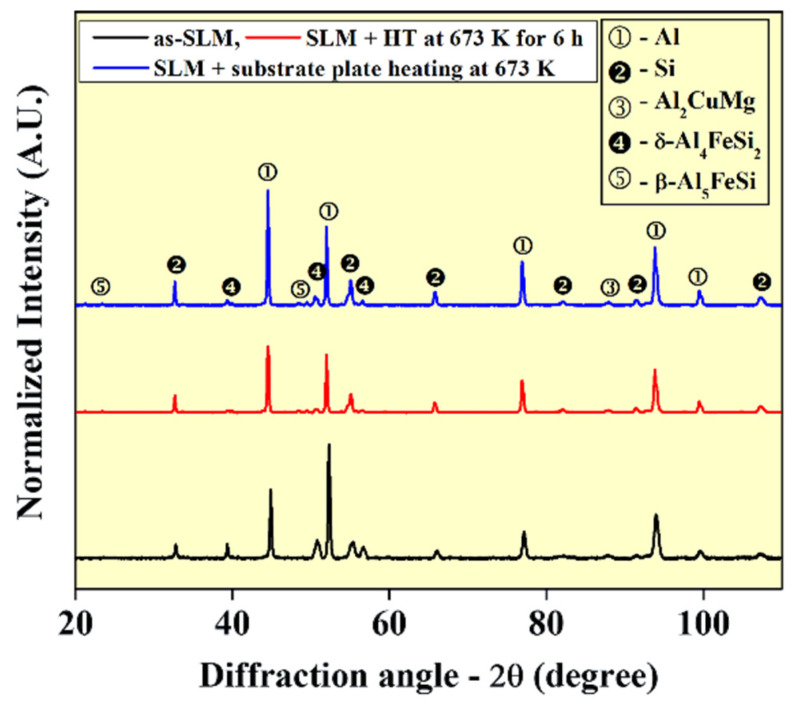
X-ray diffraction patterns of the SLM processed Al-20Si-5Fe-3Cu-1Mg samples.

**Figure 5 materials-14-00330-f005:**
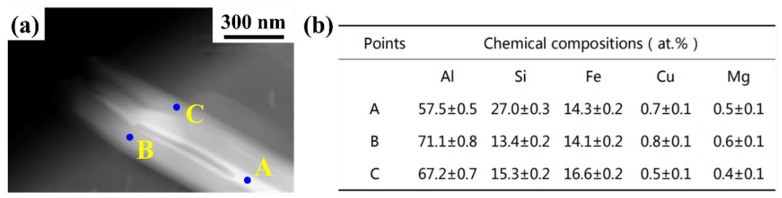
Transmission electron microscopy bright field image (**a**) showing Al-Fe-Si intermetallic phase in the SLM fabricated Al-20Si-5Fe-3Cu-1Mg sample with substrate plate heating and (**b**) table showing the EDS analysis data observed for the points marked in (**a**).

**Figure 6 materials-14-00330-f006:**
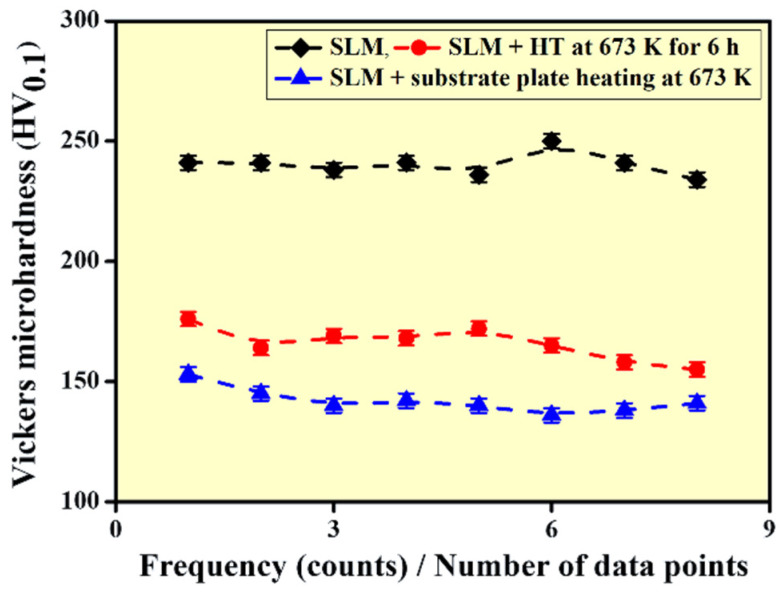
Vickers microhardness data of the Al-20Si-5Fe-3Cu-1Mg alloy observed under different conditions namely: as-prepared SLM, SLM + thermal treatment at 673 K for 6 h and finally SLM fabrication with a substrate plate heating of 673 K.

**Figure 7 materials-14-00330-f007:**

Hardness contour maps for the Al-20Si-5Fe-3Cu-1Mg SLM sample plotted from the nano-indentation data: (**a**) as-prepared SLM, (**b**) SLM + thermal treatment at 673 K for 6 h, and (**c**) SLM with substrate plate heating at 673 K.

**Figure 8 materials-14-00330-f008:**
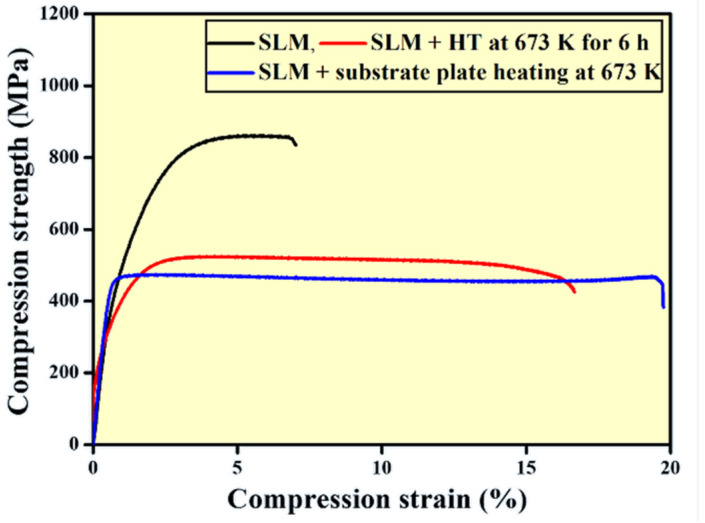
Compressive stress-strain plot of the Al-20Si-5Fe-3Cu-1Mg alloy at room temperature in the as-prepared SLM, SLM + heat treatment at 673 K for 6 h, and SLM with substrate plate heating at 673 K.

**Table 1 materials-14-00330-t001:** Chemical composition of gas atomized powders (wt%).

Elements	Si	Fe	Cu	Mg	Al
Al-20Si-5Fe-3Cu-1Mg	21.47	4.73	2.50	0.90	Balance

## Data Availability

The datasets generated during and/or analyzed during the current study are available from the corresponding author on reasonable request.

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
