# Peer review of "Effect of Substrate Plate Heating on the Microstructure and Properties of Selective Laser Melted Al-20Si-5Fe-3Cu-1Mg Alloy"

_materials, 2021, doi:10.3390/ma14020330_

Round 1

Reviewer 1 Report

The report may improve if the challenges of printing Al alloys were discussed a bit more extensively. For instance, among the main structural materials (steel, Ti, Ni and Al alloys) Al has by far the largest volumetric change during solidification (~ 6-14% as compared to 1-2 for most steels, Ti and Ni alloys), which is one of the reasons for high residual stresses. Additionally as Al is one of the most reflective materials (next to Ag), which means that 90% of the laser energy dissipate into the powder bed and / or the liquid creating additional challenges for microstructure uniformity (=oversimplification). Using pre-sintering techniques (as are commonly employed when printing Ti-Al) to change the laser interaction with a lose powder bed to a an interaction with a compact does not work either as Al generally does not sinter unless an oxide removal agent is used. The stable Al oxide, whether it evaporates, or gets somehow incorporated also poses a challenge as it obviously creates microstructure inhomogeneities. E- beam techniques are really not a good alternative as they require vacuum under which Al tends to evaporate. 

In short there are significant challenges when printing Al alloys, that are not encountered when printing other alloy systems. Using a heated substrate provides one option to address some of the challenges. They may not become that obvious, when only simple compression coupons are printed, but certainly become serious, when printing real structures. The report would gain value if in the introduction the authors were to discuss and elaborate on some of said challenges, it would provide a good reason why to investigate the effect of a heated substrate. 

Author Response

We thank the reviewer for the comments. As suggested by the reviewer, the introduction part was re-organized to highlight the challenges and the reasons pointing out the use of a heated substrate plate.

Reviewer 2 Report

Review of paper no. materials-1064711 titled Effect of substrate plate heating on the microstructure and properties of selective laser melted Al-20Si-5Fe-3Cu-1Mg alloy by P. Ma et al.

This paper investigates the effect of substrate plate heating on microstructure and mechanical properties of selective laser melted aluminum alloy. A uniform microstructure was achieved. Furthermore, a plastic deformation was slightly improved. The paper brings in new results. It is well-prepared and thoroughly discussed. I recommend accepting this paper for publication subject to minor revision.

1.The paper is based on a previous work by the authors (P. Ma et al., Microstructure and phase formation in Al-20Si-5Fe-3Cu-1Mg synthesized by selective laser melting, Journal of Alloys and Compounds 657 (2016) 430-435). Therefore, I would recommend to highlight the difference in thermal treatment between SLM and substrate plate heating. Please, provide a scheme of the selective laser beam melting and explain the substrate plate heating.

2.Present and past tenses should not be mixed in materials and methods section. This section should be written in the past tense unanimously as it relates to the experiments you did in the past.

3.Please, provide the wavelength of your X-ray beam (CoKα radiation, line 108).

4.The peritectic transformation between δ-Al4FeSi2 and β-Al5FeSi should be expressed by equation (line 242).

End of comments

Author Response

Reviewer 1

Review of paper no. materials-1064711 titled Effect of substrate plate heating on the microstructure and properties of selective laser melted Al-20Si-5Fe-3Cu-1Mg alloy by P. Ma et al.

This paper investigates the effect of substrate plate heating on microstructure and mechanical properties of selective laser melted aluminum alloy. A uniform microstructure was achieved. Furthermore, a plastic deformation was slightly improved. The paper brings in new results. It is well-prepared and thoroughly discussed. I recommend accepting this paper for publication subject to minor revision.

Thanks for the comments. The manuscript has been modified as suggested. Hope you may find the modified version acceptable for publication.

  1. The paper is based on a previous work by the authors (P. Ma et al., Microstructure and phase formation in Al-20Si-5Fe-3Cu-1Mg synthesized by selective laser melting, Journal of Alloys and Compounds 657 (2016) 430-435). Therefore, I would recommend to highlight the difference in thermal treatment between SLM and substrate plate heating. Please, provide a scheme of the selective laser beam melting and explain the substrate plate heating.

As suggested by the author the difference between the external heat treatment and the substrate plate heating is introduced in the revised version of the manuscript. The following sentence has been introduced in the introduction section, which is as follows: ‘Generally, samples by SLM are fabricated over a base plate/substrate plate. The heat extraction from the melt pool happens through the substrate plate, which is solid material made of the same composition as the powder or closest matching composition. Hence, the physical properties of the solidified melt pool and the substrate plate are similar.

  1. Present and past tenses should not be mixed in materials and methods section. This section should be written in the past tense unanimously as it relates to the experiments you did in the past.

Thanks for the comments. The materials and methods section is unanimously written in past tense.

  1. Please, provide the wavelength of your X-ray beam (CoKα radiation, line 108).

As suggested, the wavelength of the X-ray beam is introduced. The sentence is modified as: ‘….(λ = 1.7902 Å)….’

  1. The peritectic transformation between δ-Al4FeSi2 and β-Al5FeSi should be expressed by equation (line 242).

We apologize for the mistake and the transformation is not a peritectic one and has been removed accordingly.

Reviewer 3 Report

Dear Authors,

The topic of the article is novel and the findings add new knowledge to the field of the SLM of aluminium alloys. The article is clearly written and logically structured, but I recommend its publication after minor revision. Please, find my comments below.

General comments:

An extensive and detailed discussion of the results, extended with an analysis of the literature, but most of which was published without the last 3 years. Unfortunately, there are few publications from Metals, Materials, Applied Science or other MDPI Journals in the literature review.

Other comments:

1) Row 87: How was the chemical composition analysed?

2) Row 91: How was the oxygen content measured? What was the argon flow rate?

3) Data from rows 94-96 would be more advantageously presented in a table.

4) Row 102: I suggest using light microscopy instead of optical microscopy.

5) Row 121: I suggest using the force in newtons.

6) The general view of the samples after 3D printing is missing.

7) Row 328: The reference to the hardness of the powder is doubtful despite the result corresponding to the hardness of the sample after SLM (the diagonal of the indentation at 234HV hardness under the load of HV0.1 is ~ 28 um, while the particle size is 25-38 um...).

8) Row 354: What is the purpose of showing the chart? Were the measurements made in-line? What were the distances between the indentation? Where do the error bars for each measurement come from? Have more than one line been made?

9) The data presented in rows 237-239, 250-251, 257-260, 300-302, 395-402 require a detailed presentation of the results or reference.

10) Sentence in rows 385-387 to be redrafted, "smelting" in row 48 for improvement and space before the unit to insert in row 144.

Author Response

Reviewer 2

The topic of the article is novel and the findings add new knowledge to the field of the SLM of aluminium alloys. The article is clearly written and logically structured, but I recommend its publication after minor revision. Please, find my comments below.

General comments:

An extensive and detailed discussion of the results, extended with an analysis of the literature, but most of which was published without the last 3 years. Unfortunately, there are few publications from Metals, Materials, Applied Science or other MDPI Journals in the literature review.

Thanks for the comments. Unfortunately not many publications were observed for Al-20Si-5Fe-3Cu-1Mg alloy that too in the last three years. Hence, necessary references are suitably included in the manuscript.

Other comments:

1) Row 87: How was the chemical composition analysed?

Chemical composition was analysed using spectroscopy method and is introduced in the revised version.

2) Row 91: How was the oxygen content measured? What was the argon flow rate?

The oxygen level was measured using the inbuilt oxygen sensor present in the SLM device. The argon flow rate was observed as 2.5 l/min.

3) Data from rows 94-96 would be more advantageously presented in a table.

We may partially agree with the reviewers comments. However, we are not comparing the present process parameters with another alloy here and so presenting in the form of a table may highlight the data/parameters but occupy unwanted space. Hence we refrain from introducing a table for the process parameters.

4) Row 102: I suggest using light microscopy instead of optical microscopy.

As suggested, light microscope replaces optical microscope in the revised version of the manuscript.

5) Row 121: I suggest using the force in newtons.

As suggested, the force was expressed in newtons.

6) The general view of the samples after 3D printing is missing.

The general view of Al-20Si-5Fe-3Cu-1Mg sample after SLM is already published and hence it is not repeated here but cited.

7) Row 328: The reference to the hardness of the powder is doubtful despite the result corresponding to the hardness of the sample after SLM (the diagonal of the indentation at 234HV hardness under the load of HV0.1 is ~ 28 um, while the particle size is 25-38 um...).

As mentioned in the experimental part, the particle size distribution is observed to be 20-63 µm. For the hardness measurements, powder particles ranging between 40-63 µm was chosen to avoid misinterpretation of results at the load considered.

8) Row 354: What is the purpose of showing the chart? Were the measurements made in-line? What were the distances between the indentation? Where do the error bars for each measurement come from? Have more than one line been made?

The purpose of showing the hardness values in the form of a chart is to show the hardness distribution with a short range (line). Yes, the measurements were made in the form of lines. The distance between the indentations were ~100 µm so that there will be no influence between two indents. More than one line were measured for each sample so that error bars were calculated.

9) The data presented in rows 237-239, 250-251, 257-260, 300-302, 395-402 require a detailed presentation of the results or reference.

As suggested, a detailed explanation/references were introduced where necessary.

10) Sentence in rows 385-387 to be redrafted, "smelting" in row 48 for improvement and space before the unit to insert in row 144.

Typos, format and English corrections were carefully carried out and errors were rectified.

Round 2

Reviewer 3 Report

Dear Authors,

Thank You for Your answer. I still have some doubts:

Row 93: replace (OM) to (LM).

Ad. 6. Please forgive me, but in the only newly marked yellow source [11] in the text (row 137), there is no general view of the sample.

Ad. 7. Please forgive me, but the hardness of 234 HV from source [59] refers to the range of 25-38 µm. If the comparison is to concern the range of 40-63 µm, then in the source [59] the adequate hardness is 228 HV for the range of 38-63 µm.

Ad. 8. Information about the distance between the indentation should appear in the text of the manuscript.

Ad. 9. In my opinion the corrections are insufficient. For example, the reviewer is not able to judge whether the dislocation density has been correctly determined.

Author Response

Row 93: replace (OM) to (LM).

OM has been replaced with LM as suggested by the reviewer.

Ad. 6. Please forgive me, but in the only newly marked yellow source [11] in the text (row 137), there is no general view of the sample.

The general view of the sample is explained in the first paragraph of the results and discussion section between lines 134-149. We do not want to elaborate because some of the general features were already published and the report is suitably cited. Hence only the necessary general information is provided herewith.

Ad. 7. Please forgive me, but the hardness of 234 HV from the source [59] refers to the range of 25-38 µm. If the comparison is to concern the range of 40-63 µm, then in the source [59] the adequate hardness is 228 HV for the range of 38-63 µm.

Yes, there is a slight difference between the hardness between 228 – 234 HV. However, within the experimental limits, the difference is not significant and so we have said in the manuscript that the hardness are close to each other. Moreover, we are comparing the hardness of the powder, SLM sample and the sample fabricated by different consolidation method with different cooling rates (from the source [59]). However, the hardness were found to be similar around 230 HV, which is the intention of the discussion.

Ad. 8. Information about the distance between the indentation should appear in the text of the manuscript.

As suggested by the reviewer the distance between the indentations is introduced in the text (see experimental section).

Ad. 9. In my opinion the corrections are insufficient. For example, the reviewer is not able to judge whether the dislocation density has been correctly determined.

The equation used for the dislocation density calculation is introduced in the experimental section. The addition is suitably highlighted.